# Genetic Localization and Homologous Genes Mining for Barley Grain Size

**DOI:** 10.3390/ijms24054932

**Published:** 2023-03-03

**Authors:** Yi Hong, Mengna Zhang, Rugen Xu

**Affiliations:** 1Key Laboratory of Plant Functional Genomics of the Ministry of Education, Yangzhou University, Yangzhou 225127, China; 2Jiangsu Key Laboratory of Crop Genomics and Molecular Breeding, Yangzhou University, Yangzhou 225127, China; 3Jiangsu Co-Innovation Center for Modern Production Technology of Grain Crops, Yangzhou University, Yangzhou 225127, China; 4Joint International Research Laboratory of Agriculture and Agri-Product Safety of Ministry of Education of China, Yangzhou University, Yangzhou 225009, China

**Keywords:** barley, grain size, yield, QTL hotspot, homolog

## Abstract

Grain size is an important agronomic trait determining barley yield and quality. An increasing number of QTLs (quantitative trait loci) for grain size have been reported due to the improvement in genome sequencing and mapping. Elucidating the molecular mechanisms underpinning barley grain size is vital for producing elite cultivars and accelerating breeding processes. In this review, we summarize the achievements in the molecular mapping of barley grain size over the past two decades, highlighting the results of QTL linkage analysis and genome-wide association studies. We discuss the QTL hotspots and predict candidate genes in detail. Moreover, reported homologs that determine the seed size clustered into several signaling pathways in model plants are also listed, providing the theoretical basis for mining genetic resources and regulatory networks of barley grain size.

## 1. Introduction

Barley (*Hordeum vulgare* L.) is one of the earliest domesticated crops, which has a variety of uses and wide adaptability in agriculture [1]. The introduction of semi-dwarf alleles (e.g., *uzu*, *sdw1*/*denso*) increased barley lodging resistance and fertilizer utilization in agricultural production, leading to a large increase in grain yield [2,3,4]. Thanks to the Green Revolution, the global barley yield performance has increased from 1328.2 kg/ha to 2975.5 kg/ha during the past 60 years with an overall increase of 124.02% (FAO 2022, https://www.fao.org/faostat/en/ (accessed on 26 February 2023)). However, with the harsh climatic conditions, excessive exploitation, and stalling breeding process, the barley yield will be threatened [5,6]. How to improve yield to meet the increasing global food demand remains a key issue.

Grain size is a desirable alternative target trait that is closely related to yield and quality. With the technical advances in high-throughput sequencing, positional cloning of the genes regulating grain size has been sequentially reported in monocot and dicot plants including rice (*Oryza sativa*) [7,8], wheat (*Triticum aestivum*) [9], maize (*Zea mays*) [10], *Arabidopsis* (*Arabidopsis thaliana*) [11,12], and soybean (*Glycine max*) [13]. By literally studying genes mostly reported in the model plants *Arabidopsis* and rice, multiple genetic pathways regulating grain sizes have been proposed. These include the ubiquitin-proteasome pathway, mitogen-activated protein kinase signaling, G protein signaling, phytohormone signaling pathway, HAIKU pathway, and transcriptional regulators [14].

Apart from the Green Revolution genes (e.g., *sdw1*, *uzu1.a*) [15,16], row-number genes (e.g., *vrs1*, *Int-c*) [17,18] and the naked gene (*nud*) [19] are also associated with grain size. However, not a single gene in barley associated with grain size has been cloned yet via a map-based cloning technique. This is not only due to its diploid nature, but also because of the large and complex genome in barley, making it more difficult to complete sequence parse and gene annotation. In 2016, the first barley reference genome was released, which offered entry level access for genomic research given that a large number of bases are unknown [20]. Subsequently, the reference genome has been improved and updated, along with the recent release of the barley pan-genome [21,22,23]. Highly abundant repetitive elements in barley genome raise difficulties in assembly, affecting the integrity of the reference genome. Not only is the huge genome size affecting gene mapping and cloning in barley, but also the technical difficulties that sit as bottlenecks in gene functional verification. For instance, compared to rice, suitable materials that can be efficiently used for barley genetic transformation have thus far been very limited. To the best of our knowledge, only the Scottish malting barley cultivar Golden Promise has been recognized as the most efficient genotype for genetic transformation given its best shoot recovery from callus [24,25].

Marker-based mapping approaches such as QTL linkage analysis and whole-genome association studies are the current mainstream methods for identifying the preliminary QTL of grain size in barley. A segregating population used for linkage analysis usually requires bi-parents with significant differences in target traits for better QTL identifications [26]. However, the construction process is time-consuming and labor-intensive due to its specificity. The whole-genome association analysis is based on linkage disequilibrium [27], so the population employed in such an approach requires abundant genetic diversity. Combined with the algorithm, we can detect more loci associated with target traits including rare variations and minor-effect loci.

Furthermore, homology-based cloning and transcriptomic analyses are also performed for gene mining. In general, more accurate and reliable localization results can be achieved through combinations of multiple methods. For example, a major QTL for kernel length–width ratio was identified using QTL mapping and further validated through bulked segregant analysis in wheat [28]. The genetic architecture of the maize kernel size was characterized by the combination of association and linkage mapping [29]. The *P1^pet^* locus with pleiotropic effects on the spike and grain-related traits was fine mapped at a genomic interval of less than 1 Mb using linkage analysis, and further RNA-Seq was performed to predict the most possible candidate gene [30].

In this review, we summarized the achievements in the barley grain size research with an emphasis on the results of QTL linkage analysis and genome-wide association studies. Our discussion covers several important QTL hotspots and candidate or homologous genes that have been reported to function on grain size. Additionally, we also list a number of barley homologs from rice, wheat, maize, and *Arabidopsis*, which offers a theoretical basis for homology-based cloning and molecular mechanism studies.

## 2. Characteristics of Barley Grain Size

Grain size is one of the yield components with high heritability [31,32], which is closely related to grain shape and weight. Before grain filling, the spikelet hull has already developed and set the volume of the cavity within which the integuments formed the seed coat after fertilization [14]. Both spikelet hull and seed coat affect the final shape of the barley grain. Grain filling is mainly a process of assimilate accumulation, and saccharides, proteins, and lipids are the three primary storage substances accounting for the total dry matter weight [33]. This is a key stage that determines the final grain weight and yield.

The grain size affects not only the yield, but also the quality [34]. Although the quality requirements of malting barley vary among different industry standards (in different countries), the bulk density or thousand-grain weight, proportion of screenings (<2.5 mm), and protein content are all important evaluation metrics [35]. These physicochemical properties reflect the uniformity and commercial value of barley grain, which are closely associated with length, width, plumpness, and weight. A plumper and more uniform grain is preferred to ensure consistent processing and high malt extract yields [17]. Thus, genetic regions associated with grain size tend to coincide with those associated with the malt extract [36].

## 3. QTL Mapping and Association Studies on Barley Grain Size

During the past 20 years, QTL mapping of barley grain size has produced ample results. The loci controlling grain size was distributed on all seven chromosomes including ~200 QTLs and ~270 MTAs (marker-trait associations) obtained through linkage analysis and whole-genome wide association analysis, respectively. A considerable number of these have been co-detected in multiple studies due to overlapping regions and/or inter-trait correlations. We condensed 78 QTLs and 31 MTAs into 14 QTL hotspots on seven chromosomes (Table 1). Identification of the QTL hotspots was based on the physical positions of the QTLs/MTAs previously reported during the last two decades. Generally, the physical map position of QTLs/MTAs is usually disclosed in published research cases. However, if not, genetic markers associated with the QTLs/MTAs were then used for online blasting against the barley reference genome (Marker (ipk-gatersleben.de)), which identifies the corresponding physical position on the chromosomes. Thresholds were followed when deciding the QTL hotspot: one QTL locus was repeatedly identified by over three independent research cases (overlapped physiological position exist), subsequently, the total spanning physical distance by these QTLs/MTAs was regarded as one QTL hotspot. The QTL hotspots on various chromosomes will be discussed in conjunction with a review of genes related to grain yield in barley and the homologs controlling the seed size, which are reported in other plants (Table 2).

### 3.1. QTL Hotspots on Chromosome 1H

Based on multiple studies, we identified three QTL hotspots on chromosome 1H involving ten QTLs and six MTAs [5,6,36,37]. There was no QTL hotspot near the centromeric region within which high LD suppresses recombination frequencies. Located in 19–38 Mb, the QTL hotspot1H-1 contains three QTLs and two MTAs from four different studies. Among these, qSL1.1 explained the highest phenotypic variation of 16.40%, which was identified from a panel of BC3-DH lines derived from a cross between Brenda and HS584 [37]. There were two candidate genes in this region, namely, *HvGSN1* and *HvRSR1*. *HvGSN1* is an ortholog of rice *GSN1* (*GRAIN SIZE AND NUMBER1*). *GSN1* encodes the mitogen-activated protein kinase phosphatase *OsMKP1* and negatively regulates the seed size in rice [47]. *HvRSR1* is orthologous to rice *RSR1*, an APETALA2/ethylene-responsive element binding protein family transcription factor. *RSR1* indirectly affects the seed size and quality by negatively regulating the expression of type I starch synthesis genes to alter the starch component and fine structure in rice [48]. Having a content ranging from 50.5% to 75.5% in barley, starch inherently affects the grain yield and starch/protein ratios, since grain filling is also a process of starch accumulation [81]. Consequently, further functional characterization of this gene may be of importance.

QTL hotspot1H-2 spans 58 Mb from 333 Mb to 391 Mb and consists of three QTLs and one MTA [5,38,39,40]. QGl.NaTx-1H explained the highest phenotypic variation of 11.90%, which was identified using a DH population derived from a cross between Naso Nijo and TX9425. Despite their lower PVE (phenotypic variation explanation) than 10%, QTL_1H-6 and SCRI_RS_141598 were responsible for more than two grain-related traits. In this hotspot, *HvCO9* is considered as one of the candidate genes that regulate flowering under short-day conditions. *HvCO9* overexpression in rice plants caused a remarkable delay in flowering, as did *Ghd7*, which affected the grain size [51]. *vrs3* encodes a histone demethylase and controls lateral spikelet development in barley, however, it is an independent recessive gene that has only been reported in two-rowed mutants [50]. We therefore concluded that *vrs3* is not the major contributor to this hotspot, but its natural variations remain unclear. Another possible source of this QTL hotspot is suggested by the region’s overlap with two rice grain size genes, *OsBSK2* and *OsSM1* [49,52]. The rice *OsBSK2* encodes a putative brassinosteroid-signaling kinase and positively controls the grain size. Considering that the orthologs of *OsBSK2* are extremely conserved among plants (identity >80%), this gene is suggested to be an important candidate.

As for QTL hotspot1H-3 (474–500 Mb), loci controlling the grain size and flowering time are located in this region, which overlap the photoperiod gene *PPD-H2/HvFT3* [5,17,39,41,42]. *PPD-H2* was considered to be one of the main loci affecting the heading date and yield of barley in numerous studies [54,55]. In general, spring barley varieties with *PPD-H2* have an earlier heading period under short-day conditions than long-day conditions, indicating its sensitivity to a short photoperiod. While *ppd-H2* is mainly distributed in winter varieties and shows relatively late maturity [56], it can be a reliable candidate for this region. Additionally, we also identified two orthologs from rice and maize co-localizing at QTL hotspot1H-3, namely, *HvSLG* and *Hvincw1* [53,57].

### 3.2. QTL Hotspots on Chromosome 2H

Chromosome 2H is the longest chromosome with numerous loci that associate with biotic and abiotic stresses. There are two QTL hotspots identified on this chromosome involving 16 QTLs and five MTAs. As for hotspot2H-1 (17–45 Mb), three QTLs (QTL_2H-2, QTL_2H-3, and QTL_2H-4) with high LOD values but low PVE displayed pleiotropic effects and affected the multiple grain traits as described by Sharma et al. [5]. BK_12 and QTL-GL1 are associated with grain length and both showed relatively high LOD values and PVE [41,43]. In this region, another photoperiod gene *PPD-H1* can be considered as a candidate gene. *PPD-H1* is a major gene affecting the heading date under long-day conditions in barley and has significant effects on agronomic traits including yield components [59,60]. *HvSDG725*, an ortholog of rice *SDG725*, which encodes a H3K36 methyltransferase, is also located in this region. *SDG725* plays an important role in rice plant growth and wide-ranging defects occur when *SDG725* is downregulated including dwarfism, small seeds, shortened internodes, and erect leaves [58].

In hotspot2H-2, the grain size QTLs and MTAs detected in all studies overlapped in this region when the populations consisted of different row-type barleys [6,17,31,39,40,44]. A candidate gene for this hotspot is *VRS1*, which specifically expresses in lateral spikelets and inhibits their development. Wild and two-rowed cultivated barleys carry the *VRS1* while six-rowed barley carries its recessive allele and has fertile lateral spikelets [61]. Despite the number of grains in six-rowed barley being greater, there were generally fewer assimilates accumulated in a single grain and, as a result, a smaller grain size than the two-rowed barley.

### 3.3. QTL Hotspots on Chromosome 3H

Three hotspots are located in between 0–58 Mb, 454–484 Mb, and 562–565 Mb on chromosome 3H, respectively, where 19 QTLs and five MTAs were identified [5,32,36,37,39,40,41]. For hotspot3H-1, at least three candidate genes have been inferred, namely, *HvGI*, *vrs4*, and *HvCKX2*. *HvGI*, an ortholog of *Arabidopsis GIGANTEA*, participates in multiple processes from developmental regulation to physiological metabolism in plants [64]. *vrs4* was identified from six-rowed mutants with lateral spikelet fertility and loss of determinacy. Despite this, *vrs4* and *vrs3* may account for the within-sample variation of grain size, as they were all derived from induced mutants and their roles in natural variations remain unknown [62]. *HvCKX2* is orthologous to the rice *Gn1a/OsCKX2* gene, which encodes a cytokinin oxidase. It has been identified as a major contributor to grain yield improvement in rice breeding practice [76].

*uzu* and *sdw1*/*denso* are two important candidate genes for QTL hotspot3H-2 and hotspot3H-3, respectively. Semi-dwarf breeding improves the lodging resistance and fertilizer utilization of crops, leading to the enhanced yield. In barley, *sdw1/denso* and *uzu* have been designated as Green Revolution genes and possess pleiotropic effects. *sdw1/denso* encodes a gibberellic acid 20 oxidase enzyme, which is orthologous to *sd1*, and has negative effects on grain weight and quality [65]. Notably, there was no evidence that *sd1* was directly involved in the regulation of grain shape in rice while QTL intervals that overlapped with the *sdw1/denso* gene were detected to be closely associated with grain area, grain length, grain width, and grain diameter in barley [6,31].

*uzu* encodes a BR-receptor protein that is orthologous to *D61*. The BR-insensitive mutants formed small and short grains in the model plants (*Arabidopsis* and rice) and the *uzu* also reduced the grain weight by 18.8% in barley, suggesting that the phytohormone brassinosteroid plays an important role in regulating grain development [16,66,82]. In recent research, a major QTL (QGl.NaTx-3H) for grain length was identified near the *uzu* and explained 6.8–29.8% of the phenotypic variation, and this QTL showed a linkage to *uzu* but not due to gene pleiotropy [38]. However, cv.TX9425, a semi-dwarf variety carrying *uzu* used in this study, exhibited a similar small-grain phenotype as BR-insensitive mutants. Therefore, sufficiently strong recombination events are required to demonstrate whether such co-detection is due to the linkage drag or a novel locus controlling grain size.

### 3.4. QTL Hotspots on Chromosome 4H

There is a hotspot consisting of seven stable QTLs and four MTAs on chromosome 4H. The hotspot4H spans 34 Mb from 6 to 40 Mb and contains seven candidate genes. *Vrn-H2* is one of the three vernalization genes in barley that affects heading and flowering [71]. In breeding practice, the selection and utilization of different allelic combinations of vernalization and photoperiod genes can directly affect the grain yield. Another gene of interest involved in spike morphology is *INT-C*, an ortholog of the maize *TB1*. *INT-C* and *VRS1* are functionally opposed and show effects interaction, that is, the dominant *VRS1* inhibits the development of lateral spikelets, while *INT-C* promotes fertile spikelets [18]. Since grains from central spikelets are generally expected to be larger and more symmetrical than those from lateral spikelets, *INT-C* may be an essential factor affecting the grain uniformity in six-rowed barley.

Five orthologs from rice (*HvRGB1*), maize (*HvDek35*, *Hvemp4*), and *Arabidopsis* (*HvAHKs*, *HvDAR1*) were identified in hotspot4H. *RGB1* encodes the β-subunit (Gβ) of heterotrimeric G protein. Loss of function and suppression of Gβ result in short seeds in rice, suggesting that Gβ positively regulates the seed length [67]. *Dek35* and *emp4* mutants with developmental deficiency confer a seed-lethal phenotype in maize [68,70]. Genetic analysis indicated that cytokinin-dependent endospermal and/or maternal control can affect embryo size. Three histidine kinases perceive the cytokinin signal in *Arabidopsis*: AHK2, AHK3, and CRE1/AHK4 [69]. *DA1* affects the seed size in the maternal control by regulating cell proliferation in the integuments redundantly with *DAR1* in *Arabidopsis* [11].

### 3.5. QTL Hotspots on Chromosome 5H

Chromosome 5H shows significance across almost all grain-related traits (grain length, width, thickness, plumpness, and thousand-grain weight). A total of 16 QTLs and nine MTAs are clustered into three hotspots [5,6,36,39,40,42,43]. As for hotspot5H-1 (0–23 Mb), *HvIKU2* and *HvPPKL3* are two putative candidates. In *Arabidopsis*, *IKU2* functions zygotically to control the seed size by affecting endosperm development [72]. *PPKL3* encodes a protein phosphatase with the Kelch-like repeat domain, and the T-DNA insertion mutants displayed a longer grain phenotype in rice [73].

*Hvdep1*, a noncanonical Gγ of G protein, is a causal gene for another semi-dwarf locus (ari-e) in barley as described by Wendt et al. [75], which is located in hotspot5H-2 (427–431 Mb). The overexpression and downregulation of *DEP1* result in larger and smaller grains in rice, respectively [74]. Genetic transformation also demonstrated that *HvDEP1* positively regulates grain size and culm elongation in barley [75].

The QTL hotspot5H-3 (541–588 Mb) consists of seven QTLs and five MTAs and displays associations for all grain-related traits. Among these, QTL-GT2 (grain thickness) and QTL-GP2 (grain plumpness) were two consensus QTLs identified using a DH population derived from the cross Vlamingh × Buloke. There was a high PVE of 15.3% for QTL-GP2, and the linkage maker 8682-406 can be used to screen well-filled varieties in maker assisted breeding. In this region, there are three orthologs from rice (*HvDST* and *HvSK41*) and *Arabidopsis* (*HvABA2*). Rice zinc finger protein *DST* associated with abiotic stresses regulates *CKX2* expression to enhance grain production [76]. *OsSK41* is responsible for a major QTL that controls the grain size and weight in rice [83]. Abscisic acid was reported to control the seed size by regulating the HAIKU pathway, and seeds from ABA-deficient mutants exhibited increased size, mass, and embryo cellularity in *Arabidopsis* [77].

### 3.6. QTL Hotspots on Chromosome 6H

A total of seven QTLs and one MTA overlapped an interval of 31 Mb from 463 to 494 Mb for all grain-related traits on chromosome 6H [5,17,32,37,43]. Of these QTLs, qTGW6.1 explained the highest phenotypic variation ranging from 19.60% to 38.30% in multiple environments [37]. The linkage maker of this major QTL can also be used for marker-assisted selection. *HvDEK1*, an ortholog of maize *DEK1*, encodes a Calpain-Type Cysteine Protease and is located within this hotspot. In recent years, numerous studies have indicated that *DEK1* is important for the development and mechanical stimulation of seeds, leaves, and flowers. Kernels from maize *dek1* mutants are small and lacking plumpness, and deeper investigations showed that the normal *DEK1* gene products are required for aleurone cell fate specification [78,84]. Another candidate gene is *LEC1*, which encodes a transcriptional factor that regulates seed development. Mutation in the *LEC1* gene not only alters the normal developmental rhythm and pattern, resulting in abnormal embryos, but also affects the accumulation of storage substances in *Arabidopsis* and maize [80,85].

### 3.7. QTL Hotspots on Chromosome 7H

There are many QTLs and MTAs across the entire 7H chromosome, but most of these occupy separate positions according to multiple studies. We clustered four consensus QTLs and one MTA within this chromosomal interval from 519 Mb to 540 Mb into a QTL hotspot, where the *Nud* gene is a candidate for grain size [6,19,31,40,41]. Similar to *VRS1* and *INT-C*, this region can be detected in almost all studies if the populations consist of naked and hulled barley. With the deletion of *Nud*, naked barley is produced free-threshing after maturity, resulting in altered grain dimension and weight compared to hulled barley. Previous studies have also reported that yield-related QTLs are tightly linked to the *nud* gene [86,87].

### 3.8. Interrelationships of QTL Hotspots

As above-mentioned, numerous mapping experiments revealed multiple QTL hotspots that control the barley grain size on seven chromosomes. According to these candidate genes, several QTL hotspots are found to share common features, especially containing phytohormone-related genes, indicating potential interactions with each other.

Plant hormones play important roles in seed formation and can affect the grain size directly or indirectly. Among the 14 QTL hotspots we summarized, eight contained candidate genes related to plant hormones. Brassinosteroid (BR) is one of the hormones essential for plant height, spike architecture, and organ size. *BSK2* (Hotspot1H-2) and *SLG* (Hotspot1H-3) affect the seed size by altering the length of the epidermal cells of the spikelet hull through cell expansion in rice [49,53]. *BSK2* interacts directly with *BRI1* (hotspot3H-2) and affects grain size independent of the BR signaling pathway, while *SLG* is involved in BR homeostasis by positively regulating endogenous BR levels. *SDG725* (hotspot2H-1) can modulate brassinosteroid-related gene expression through epigenetic regulation including *BRI1* to affect plant growth and development in rice [58]. Cytokinin (CK) is another key regulator of plant growth and cytokinin oxidase/dehydrogenases (CKXs) catalyze CK degradation irreversibly. In previous studies, the *iku2-2* (hotspot5H-1) seed size phenotype can be partially restored by overexpressing *CKX2* (hotspot3H-1) in *Arabidopsis* [63]. Moreover, the DST-directed (hotspot5H-3) expression of rice *CKX2* affects CK accumulation in the shoot apical meristem, which controls the reproductive organ number [76].

## 4. Homologous Gene Mining of Grain Size

Grain size regulation involves a complex genetic network controlling the development of spikelet hulls, integuments, and endosperms, which are all determined components of the final grain size. Recent advances have cloned numerous genes that are involved in several networks to control the grain size including the ubiquitin-proteasome pathway, mitogen-activated protein kinase signaling, G protein signaling, phytohormone signaling pathway, HAIKU pathway, and transcriptional regulators. Some of them also exhibit genetic interactions and integrate multiple signaling pathways. These findings not only shed new light on our understanding of molecular mechanisms, but also provide key ideas for the research of homologs in other crops. For instance, *TaGW2* and *TaTGW6-A1*, which encode E3 ubiquitin ligase and indole-3-acetic acid (IAA)-glucose hydrolase, respectively, have been cloned by comparative genomics approaches. Polymorphism and haplotype analysis indicated that they are strongly associated with grain size and weight in wheat [88,89].

Orthology, which is of great interest, paralogy, and xenology are three main subclasses of homology used to describe the evolutionary relationships between species. As more high-quality genomes are being released, whole-genome alignment (WGA) is becoming a powerful tool for gaining insights into the evolutionary scope. Despite species divergence, numerous genetic features of ancestry are retained, resulting in a high level of genome collinearity among closely related species. Gene type, order, and orientation are relatively conserved within collinear blocks [90]. Based on collinearity and gene homolog analyses, 29 candidate genes related to seed shattering were identified in Chinese wild rice [91].

To facilitate homology-based cloning, here we list 190 barley orthologs of 142 grain size genes in other plants according to the Ensemble database (http://plants.ensembl.org/index.html (accessed on 26 February 2023)) (Appendix A). Rice is responsible for 110 of them, maize for 46, *Arabidopsis* for 25, and wheat for nine. Unsurprisingly, 21 grain size genes do not create a one-to-one correspondence in barley. Some of these orthologs (e.g., *HvZM-INVINH1*) are the results of tandem gene duplication, while others (e.g., *HvMADS87*) are distributed on several chromosomes due to interspersed gene duplication. Collinearity analyses using TBtools software showed extensive genome collinearity between barley and gramineous crops, but a low degree of genome collinearity between barley and *Arabidopsis* [92]. In the 190 barley orthologs analyzed, 81 shared significant collinearity with other plants, indicating deeply conserved functions (Figure 1 and Appendix A, References [93,94,95,96,97,98,99,100,101,102,103,104,105,106,107,108,109,110,111,112,113,114,115,116,117,118,119,120,121,122,123,124,125,126,127,128,129,130,131,132,133,134,135,136,137,138,139,140,141,142,143,144,145,146,147,148,149,150,151,152,153,154,155,156,157,158,159,160,161,162,163,164,165,166,167,168,169,170,171,172,173,174,175,176,177,178,179,180,181,182,183,184,185,186,187,188,189,190,191,192,193,194,195,196,197,198,199,200,201,202,203,204,205,206,207,208,209,210,211,212] are cited in Appendix A). We finally mapped these collinearity genes to the reference genome, which can provide a theoretical basis for further studies (Figure 2).

## 5. Conclusions and Perspectives

During the past two decades, significant achievements have been witnessed in crop yield and yield components, despite the threat of both biotic and abiotic stress. Grain size with high heritability has long been a primary target of breeding, which is closely related to final yield and quality. Thanks to linkage analysis and whole-genome association analysis, hundreds of grain size QTLs were identified. However, the PVE of these QTLs varied depending on the study materials and methods. How to systematically evaluate the genetic effects of these QTLs and the potential application of linkage markers in different genetic contexts are crucial in marker-assisted breeding.

According to the published data, the cloning of barley genes has been relatively rare, making it hard to identify the regulatory networks of grain size. In the future, more efforts should be invested in the fine-mapping of these reported QTLs to isolate the causal gene. On the other hand, substantial evidence from genetics and molecular biology has suggested that large SVs (structure variations) identified from the pan-genome can cause the phenotypic variance affecting many important agronomic traits [22,212,213]. Compared to traditional SNP-based GWAS, PAV-based GWAS (presence and absence variation, PAV) enable the precise identification of trait-associated genomic regions and can complement SNP-based GWAS. In barley, therefore, taking full advantage of the published pan-genomic data to mine variations will become a new strategy.

The CRISPR/Cas9 system serves as a revolutionary technique in molecular design breeding and varietal improvement in many crops. With gene editing, target function-deficient mutants can be created rapidly and efficiently without introducing exogenous genes into the genetic background. It can directly influence gene expression at the transcriptional levels, making it a mainstream tool for gene functional validation. The combination of comparative genomics and gene-editing technology will be very practical for the study of unknown genes or orthologs, and will help constantly refine the regulatory network of grain size in barley.

## Figures and Tables

**Figure 1 ijms-24-04932-f001:**
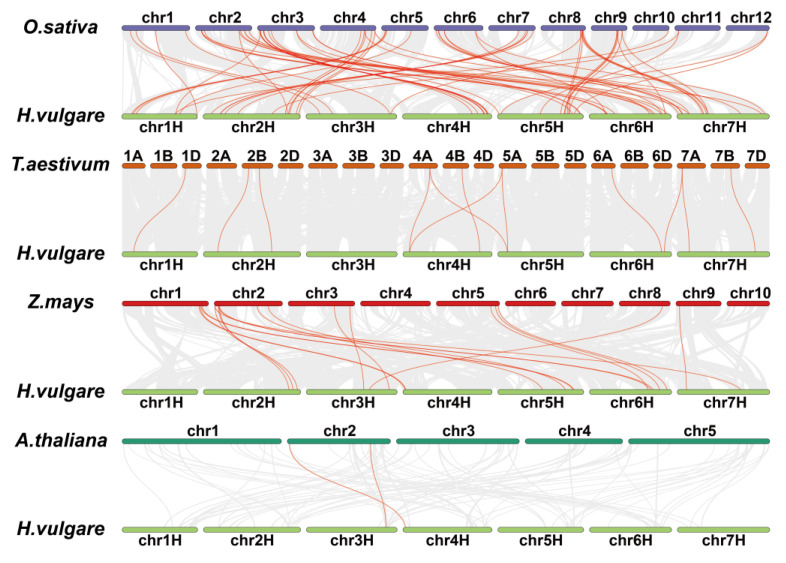
Collinearity analysis between genomes and the grain size genes of barley and other plants. Grain size genes in other plants are shown in Appendix A. The grey lines represent the collinearity of genome between *H. vulgare* and other plants, and the red lines represent the collinearity of the grain size genes between *H. vulgare* and other plants.

**Figure 2 ijms-24-04932-f002:**
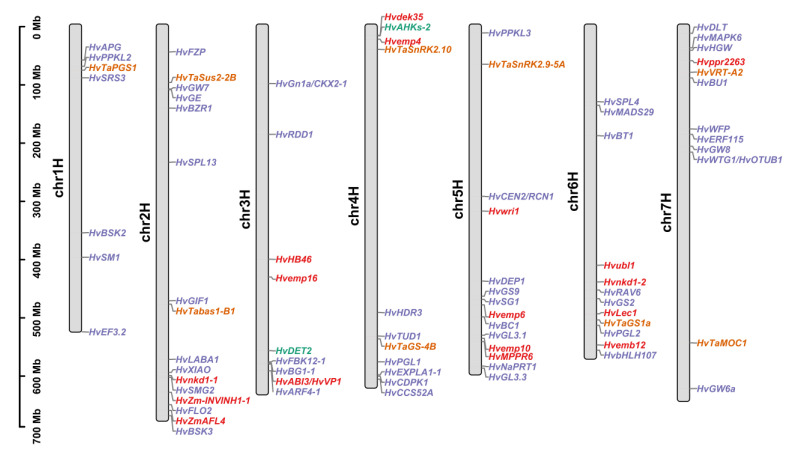
Distribution of the collinearity genes across seven barley chromosomes. The red font represents the collinearity genes from maize, the green represents the collinearity genes from *Arabidopsis*, the orange represents the collinearity genes from wheat, and the purple represents the collinearity genes from rice.

**Table 1 ijms-24-04932-t001:** Information of the grain size of the QTL hotspots across all seven chromosomes.

Chr	QTL Hotspots	Position (Mb)	QTLs or MTAs	Type	References
1H	hotspot1H-1	19–38	QTL_1H-4, qGw1-3, qSL1.1	QTL	[5,6,36,37]
			1_0186, SCRI_RS_123187	MTA	
1H	hotspot1H-2	333–391	QTL_1H-6, QGl.NaTx-1H, QTL1_TGW	QTL	[5,38,39,40]
			SCRI_RS_141598	MTA	
1H	hotspot1H-3	474–500	QTL_1H-12, QTL_1H-13, QTL_1H-14, qGL1	QTL	[5,17,39,41,42]
			SCRI_RS_188218, Cmwg706, 12_30191	MTA	
2H	hotspot2H-1	17–45	QTL-GL1, QTL_2H-2, QTL_2H-3, QTL_2H-4, QTL4_TGW	QTL	[5,40,41,43]
			BK_12	MTA	
2H	hotspot2H-2	562–583	qGL2, QTL_2H-8, QTL6_TGW, qGl2-1, qGw2-3, qTgw2-1, KW-MA-2H, cqGW2–3, cqGW2–4, QTL_2H-9, KW-BA-2H	QTL	[5,6,17,31,32,39,40,42,44]
			SCRI_RS_200291, SCRI_RS_171032, SCRI_RS_138463, vrs1	MTA	
3H	hotspot3H-1	0–58	QTL_3H-1, KW-MA-3H, QTL_3H-2, QTL_3H-3, QTL_3H-4, QTL9_TGW, qSB3.1	QTL	[5,32,37,39,40,41]
			2_0662, 12_11414, SCRI_RS_230486, SCRI_RS_115045	MTA	
3H	hotspot3H-2	454–484	KW-BA-3H.1, qGw3-5, QGl.NaTx-3H, QTL_3H-6	QTL	[5,6,32,38,39,41]
			SCRI_RS_145300, SCRI_RS_235065	MTA	
3H	hotspot3H-3	562–565	qGw3-4, QTL_3H-10, qGl3-1, cqGL3, cqGW3–1, cqGW3–2, LEN-3H	QTL	[5,6,31,45]
4H	hotspot4H	6–40	QTL_4H-1, QTL_4H-2, QTL_4H-3, QTL12_TGW, QTL_4H-4, qGL4, QTL13_TGW	QTL	[4,17,36,39,40,41]
			1_0113, SCRI_RS_180891, 12_30793, int-c	MTA	
5H	hotspot5H-1	0–23	qTgw5-1, QTL_5H-1, QTL_5H-2, QTL_5H-3	QTL	[5,6,39]
			SCRI_RS_168359, 11_10580	MTA	
5H	hotspot5H-2	427–431	QTL-GL2, qGL5H, QTL_5H-6, qTGW5, qGL5	QTL	[5,36,42,43,46]
			1_0641, 2_1239	MTA	
5H	hotspot5H-3	541–588	QTL_5H-14, QTL-GP2, QTL-GT2, QTL_5H-15, QTL_5H-16, QTL16_TGW, QTL_5H-17	QTL	[5,36,39,40,43]
			1_1071, 1_1490, SCRI_RS_159482, 11_10236, 12_30504	MTA	
6H	hotspot6H	463–494	QTL-GP3, QTL-GT3, QTL-GW2, QTL_6H-6, qTGW6.1, KW-BA-6H, KW-MA-6H	QTL	[5,17,32,37,43]
			ABC175	MTA	
7H	hotspot7H	519–540	QTL19_TGW, qGl7-1, qGw7-1, cqGL7–1	QTL	[6,31,40,41]
			12_30996	MTA	

**Table 2 ijms-24-04932-t002:** Candidate or homologous genes for the barley grain size QTL hotspots.

QTL Hotspots	Candidate Genes	Barley Gene_ID	Other Plants	Accession Number	Functional Annotation	Reference
hotspot1H-1	*HvGSN1*	HORVU.MOREX.r3.1HG0008520	Rice	Os05g0115800	Dual specificity phosphatase	[47]
	*HvRSR1*	HORVU.MOREX.r3.1HG0012250	Rice	Os05g0121600	AP2-like ethylene-responsive transcription factor	[48]
hotspot1H-2	*HvBSK2*	HORVU.MOREX.r3.1HG0052470	Rice	Os10g0571300	Serine/threonine-protein kinase BSK2	[49]
	*vrs3/int-a*	HORVU.MOREX.r3.1HG0053590	NA	NA	Lysine-specific demethylase	[50]
	*HvCO9*	HORVU.MOREX.r3.1HG0058180	NA	NA	CONSTANS-like protein	[51]
	*HvSM1*	HORVU.MOREX.r3.1HG0058550	Rice	Os05g0389000	AP2-like ethylene-responsive transcription factor	[52]
hotspot1H-3	*HvSLG*	HORVU.MOREX.r3.1HG0076820	Rice	Os08g0562500	HXXXD-type acyl-transferase family protein	[53]
	*PPD-H2/HvFT3*	HORVU.MOREX.r3.1HG0077240	NA	NA	RNA ligase/cyclic nucleotide phosphodiesterase family protein	[54,55,56]
	*Hvincw1*	HORVU.MOREX.r3.1HG0087260	Maize	Zm00001eb242820	Cell wall invertase	[57]
hotspot2H-1	*HvSDG725*	HORVU.MOREX.r3.2HG0096180	Rice	Os02g0554000	Histone-lysine N-methyltransferase	[58]
	*PPD-H1*	HORVU.MOREX.r3.2HG0107710	NA	NA	Pseudo-response regulator	[59,60]
hotspot2H-2	*VRS1/Int-d*	HORVU.MOREX.r3.2HG0184740	NA	NA	Homeobox leucine zipper protein	[61]
hotspot3H-1	*vrs4*	HORVU.MOREX.r3.3HG0233930	NA	NA	LOB domain protein	[62]
	*HvCKX2*	HORVU.MOREX.r3.3HG0236930	*Arabidopsis*	At2g19500	Cytokinin oxidase/dehydrogenase	[63]
	*HvGI*	HORVU.MOREX.r3.3HG0238250	*Arabidopsis*	At1g22770	Gigantea-like protein	[64]
hotspot3H-2	*uzu*	HORVU.MOREX.r3.3HG0285210	Rice	Os01g0718300	Receptor kinase	[38,65,66]
hotspot3H-3	*sdw1/denso*	HORVU.MOREX.r3.3HG0307130	Rice	Os07g0169700	Gibberellin 20 oxidase 2	[6,31]
hotspot4H	*HvRGB1*	HORVU.MOREX.r3.4HG0333750	Rice	Os03g0669100	Deoxyuridine 5′-triphosphate nucleotidohydrolase	[67]
	*INT-C/Vrs5*	HORVU.MOREX.r3.4HG0336720	Maize	Zm00001eb054440	Teosinte branched 1 protein	[18]
	*Hvdek35*	HORVU.MOREX.r3.4HG0337450	Maize	Zm00001eb055010	Pentatricopeptide repeat-containing protein	[68]
	*HvAHKs*	HORVU.MOREX.r3.4HG0337770	*Arabidopsis*	At2g01830	Histidine kinase	[69]
	*Hvemp4*	HORVU.MOREX.r3.4HG0339220	Maize	Zm00001eb055980	Pentatricopeptide repeat-containing protein	[70]
	*Vrn-H2*	HORVU.MOREX.r3.4HG0340200	NA	NA	RING finger and CHY zinc finger protein	[71]
	*HvDAR1*	HORVU.MOREX.r3.4HG0341060	*Arabidopsis*	At4g36860	Protein DA1-related 1	[11]
hotspot5H-1	*HvIKU2*	HORVU.MOREX.r3.5HG0421310	*Arabidopsis*	At3g19700	Receptor protein kinase, putative	[72]
	*HvPPKL3*	HORVU.MOREX.r3.5HG0426290	Rice	Os12g0617900	Serine/threonine-protein phosphatase	[73]
hotspot5H-2	*HvDEP1*	HORVU.MOREX.r3.5HG0480200	Rice	Os09g0441900	Guanine nucleotide-binding protein subunit gamma 3	[74,75]
hotspot5H-3	*HvDST*	HORVU.MOREX.r3.5HG0516880	Rice	Os03g0786400	Zinc finger protein, putative	[76]
	*HvABA2*	HORVU.MOREX.r3.5HG0524120	*Arabidopsis*	At1g52340	Short-chain dehydrogenase/reductase	[77]
hotspot6H	*Hvdek1*	HORVU.MOREX.r3.6HG0608170	Maize	Zm00001eb014030	Calpain-like protein	[78]
	*HvLec1*	HORVU.MOREX.r3.6HG0611100	Maize	Zm00001eb253260	Nuclear transcription factor Y subunit B	[79,80]
hotspot7H	*Nud*	HORVU.MOREX.r3.7HG0719680	NA	NA	Ethylene-responsive transcription factor	[19]

## Data Availability

Data sharing not applicable.

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
