# Peer review of "Genetic Localization and Homologous Genes Mining for Barley Grain Size"

_ijms, 2023, doi:10.3390/ijms24054932_

Round 1

Reviewer 1 Report

In this review, the authors summarise the progress made in barley grain size research in the last two decades, focusing on the results of QTL linkage analysis and genome-wide association studies. Several important QTL hotspots are inferred and candidate or homologous genes reported to influence grain size are discussed. In addition, barley grain size homologous genes from rice, wheat, maize and Arabidopsis are considered to contribute to gene mining of candidate genes and understanding of regulatory networks controlling barley grain traits.

The claim for why such a review is relevant is adequately addressed in the introduction. The manuscript is well structured and contains two comprehensive tables of grain size QTL hotspots and candidate genes resp. homologous genes for barley grain size within the QTL hotspots. In addition, the previously described QTL and MTAs together with the candidate genes and homologues belonging to the hotspots are well discussed in separate subchapters. Table S1, was not given to the reviewer and has so far not been reviewed together with chapter 4. Tables (1, 2) and figures (1, 2) are very relevant and helpful to understand the review.

However, a major shortcoming is that there is no mention of how the 78 QTL and 31 MTAs were reduced/condensed into QTL hotspots and how the physical location of the hotspots was inferred. This needs to be explained in detail, otherwise the hotspots will have to be believed rather than understood or verified the very best.

The reviewer also suggests that the following issues, listed below, should be addressed in order to accept the manuscript. The manuscript should revised by a professional language editing service to correct grammar and ensure that your meaning is clear and concise.

[11] “have” (instead of “has”)

[24] use common units such as “t/ha” or “kg/ha”

[24-25] data should refer to latest FAO numbers

[26] delete “excellent” and the 2nd “the”

[28-30] give reference

[46] “a large number of “N”s” is scientific slang, rephrase appropriately

[48] add that gene mapping and cloning is also affected by the abundance of repetitive elements

[55-56] rephrase the sentence and use proper scientific language

[57] rephrase as a conditional sentence

[71] add “research” after “barley grain size”

[78] give reference for high heritability of barley grain size

[82] delete “s” in “assimilates”

[83] add “ing” to “account”, delete the 2nd “the”

[99-100] explain in technical detail how the QTL and MTAs have been condensed into the hotspots and how the physical position was inferred

[108] add “lines” or adequate after “BC3-DH”

[123] introduce the abbreviation “PVE” at least once

[124-125] mention shortly how that relates to grain size

[134-135] traits are not located in the genome, please rephrase

[144] delete “d” in “associated”

[148-149] That is not a complete sentence. Please clarify.

[168] replace “exist” by “have been inferred”

[188] lowercase “A”

[190] blank after QTL

[194] add “a” after “or”

[197] delete “putative” as it is only a candidate

[200] delete “the final”

[203-204] delete the latest “the”, add “are” after “spikelets”

[211] add “in maize” after “phenotype”

[225] give the reference according to the journal’s rules

[251] delete “regulatory”, add “s” to “regulate”

[253] add “s” to “affect”

[257] delete “lying”

[258] add “for grain size” after “candidate” if appropriate

[266] What is meant by “are found to share common features”? Please clarify.

[269] delete “of them”

[284] delete “and the” and replace by “controlling the”

[301] delete “The”, start the sentence with “Gene”

[307, 315] The mentioned table S1 was not given to the reviewer, thus the content of the table was not reviewed together with chapter 4.

[330] change “the” to “a”

[342] the abbreviation “PAV” should be written-out as it might not obvious to every reader

[349] replace “regulate” by “influence” and delete “s” in the 2nd “genes”

Reviewer 2 Report

To,

The Editor,

IJMS, MDPI,

Manuscript ID: ijms- 2216101

Subject: Submission of comments of the manuscript in “IJMS"

Dear Editor IJMS, MDPI,

Thank you very much for the invitation to consider a potential reviewer for the manuscript (ID: ijms- 2216101). My comments responses are furnished below as per each reviewer’s comments. 

The manuscript by Hong et al. summarize the progress made in molecular mapping of barley grain size over the past two decades, mainly on the results from QTL linkage analysis and genome-wide association studies. We discuss in detail the QTL hotspots and predict candidate genes for further research. Moreover, homologs reported to control seed size in several signaling pathways in model plants are also listed, further providing a theoretical basis for gene mining and regulatory networks of barley grain size. The aim of the work is interesting, providing important information to elucidate the molecular mechanisms of barley grain size is vital for cultivar improvement and accelerating breeding processes. however, in my opinion, the MS needs major revisions. I have some suggestions to improve this manuscript:

  1. The topic is relevant and interesting. The biggest problem I had with the paper was the poor English language, which made the text difficult to understand. The paper needs to go through extensive language editing before it can be considered.
  2. The manuscript is not consistent, I can not understand the purpose of this review.
  3. The introduction is not properly contextualized. It needs great improvement in explaining both merits and demerits of invasive species and methods of identification.
  4. In the manuscript, the scientific names of plant species should be italic.
  5. When describing the molecular pathway involved, please make it more clear in which species the molecular knowledge was obtained.
  6. The figures' quality is not up to the standards. The authors are strongly recommended to prepare good quality figures including the 1 and 2

Round 2

Reviewer 1 Report

Review of the revised manuscript by Hong Y, Zhang M, and Xu R Genetic Localization and Homologous Genes Mining for Barley Grain Size - IJMS (ijms-2216101) Section Molecular Plant Sciences - Special Issue Genetics and Genomics-Based Crop Improvement and Breeding

Dear authors

Thank you for giving full consideration to the reviewer's suggestions, including careful editing of the manuscript.

However, there remains one point with which the reviewer disagrees.

[33-35] “However, with the harsh climatic conditions, excessive exploitation and stalling breeding process, the increase of barley yield gradually reaches a plateau [5,6].”

I have checked the articles [5,6] and cannot see how the aspects of the sentence are supported by these sources. According to the FAO data, barley yield trends have plateaued or even declined in Western Europe, Africa, Eastern and Southern Asia in the last 10 years, but are still increasing worldwide (https://ourworldindata.org/grapher/barley-yields?tab=table). This is consistent with the literature I am aware of. So please give appropriate references, rewrite the sentence or limit it to certain regions or time intervals.

If this issue is addressed, I agree to publication in IJMS.

Yours sincerely

Author Response

#1[33-35] “However, with the harsh climatic conditions, excessive exploitation and stalling breeding process, the increase of barley yield gradually reaches a plateau [5,6].” I have checked the articles [5,6] and cannot see how the aspects of the sentence are supported by these sources. According to the FAO data, barley yield trends have plateaued or even declined in Western Europe, Africa, Eastern and Southern Asia in the last 10 years, but are still increasing worldwide (https://ourworldindata.org/grapher/barley-yields?tab=table). This is consistent with the literature I am aware of. So please give appropriate references, rewrite the sentence or limit it to certain regions or time intervals.

Reply: Thanks for your comments. We have rewritten this sentence as below. “However, with the harsh climatic conditions, excessive exploitation and stalling breeding process, the barley yield will be threatened [5,6]. How to improve yield to meet increasing global food demand remains a key issue.” (Line 29-31)

Reviewer 2 Report

Dear Editor,

Thank you for providing the opportunity to review the revised manuscript. The manuscript is improved considerably after revision according to the reviewer's comment. Now this study is a suitable contribution to the IJMS. I recommend the manuscript for publication.

Thank you

With best regards

Author Response

# Thank you for providing the opportunity to review the revised manuscript. The manuscript is improved considerably after revision according to the reviewer's comment. Now this study is a suitable contribution to the IJMS. I recommend the manuscript for publication.

Reply: We thank reviewer 2 for their approval of the revised manuscript.